# OpenReview forum: "TrustGen: A Platform of Dynamic Benchmarking on the Trustworthiness of Generative Foundation Models"
_ICLR.cc/2026/Conference — ICLR 2026 Poster_

### Official Review · Reviewer_u6eR · 2025-11-01

**Soundness:** 2
**Presentation:** 3
**Contribution:** 3
**Rating:** 4
**Confidence:** 3

**Summary:**

This paper presents TRUSTGEN, a dynamic and modular benchmarking platform designed to evaluate the trustworthiness of generative models. TRUSTGEN develops a dynamic evaluation pipeline encompassing three modules—the Metadata Curator, the Test Case Builder, and the Contextual Variator—to enable the automatic and iterative generation of evaluation data. The authors assessed 39 models and summarized their observations.

**Strengths:**

* The engineering effort behind this paper is impressive. The work involves the design and evaluation across dozens of trustworthiness dimensions and models. The authors provided the source code and datasets.

* The paper introduces several new, fine-grained trustworthiness dimensions, such as "Self-Doubt Sycophancy" and "Exaggerated Safety".

**Weaknesses:**

* The motivation for proposing the evaluation framework for T2I, LLM, and VLM models is not clear. While Table 3 suggests that the authors' evaluation framework covers a greater number of models and dimensions, the question remains: Why can't existing benchmarks, if combined, achieve the same purpose? What is the critical shortcoming of existing benchmarks? The potential advantage is the construction of dynamic datasets. However, this could also be a drawback, as the quality of evaluation data cannot be fully guaranteed before rigorous human checking.

* The structure of the paper is somewhat unreasonable for a benchmarking study. The main body should present the detailed information about dimension, dataset (size, case examples), and metrics. Furthermore, I note from Table 1 that the evaluation dimensions for different model types appear to be defined independently. Why should the dimensions of trustworthiness vary for each model? For instance, if LLMs can generate toxic output, why would VLMs be exempt from this concern [1]? I suggest the authors provide more rigorous definitions for their trustworthiness dimensions. By the way, safety and robustness are often discussed synonymously within the context of trustworthy AI, so their distinct roles in this work need more justification. Additionally, critical threats like backdoor attacks and watermarking are significant components of trustworthy AI, yet they are not discussed in this paper.

[1] On the trustworthiness landscape of state-of-the-art generative models: A survey and outlook

---

Note: My review is primarily based on the main body of the paper.

**Questions:**

See Weaknesses.

---

> ### Author Response · Authors · 2025-11-15
> **Response for Reviewer u6eR (Part I)**
>
> Dear Reviewer u6eR,
>
> Thank you very much for your detailed review and constructive suggestions. We truly appreciate the time you have dedicated to understanding our work, especially given the substantial length of the submission.
>
> **A note on manuscript organization:** We noticed your comment that "the review is primarily based on the main body of the paper." We completely understand this, our submission indeed exceeds 100 pages with appendices, and we recognize it can be challenging to review every section in detail under time constraints. We genuinely appreciate your effort in engaging with the main text. To help address your concerns more effectively, we want to note that some technical details and validation results were placed in specific appendix sections (particularly **Appendix H, J, and K** for data quality validation, and **Appendix D and P** for benchmark details). Below, we address each of your questions and provide pointers to relevant sections where appropriate.

---

> ### Author Response · Authors · 2025-11-15
> **Response for Reviewer u6eR (Part II)**
>
> ### Q1: Motivation for proposing the evaluation framework and advantages over combining existing benchmarks
>
> **Our Response:**
>
> We appreciate this question, as it goes to the core of our design philosophy. Our work is driven by two intertwined motivations: **to build the first dynamic trustworthiness benchmark for generative models** and to provide a unified evaluation infrastructure that can consistently assess different model types, including LLMs, VLMs, and text to image models.
>
> Regarding why combining existing benchmarks is not sufficient, the key issue is that current trustworthiness benchmarks are almost exclusively static, which leads to fundamental limitations. As discussed in §2.1 and §3.1, static benchmarks suffer from three main problems:
>
> (1) Data leakage. Once released, benchmark data rapidly propagates into training and fine tuning pipelines, leading to inflated and potentially misleading performance estimates [1, 2, 3].
>
> (2) High construction cost. Each new benchmark requires manual curation, annotation, and validation from scratch. This makes it prohibitively expensive to refresh benchmarks or expand coverage in step with rapidly evolving models.
>
> (3) Limited adaptability. Static datasets become outdated quickly and cannot capture the emerging capabilities and failure modes of modern generative models.
>
> Recent work on general model evaluation has therefore started to explore dynamic benchmarking to address these issues [4, 5, 6, 7]. However, to the best of our knowledge, this paradigm has not yet been applied to trustworthiness evaluation. Our primary motivation is to fill this gap by providing a **dynamic trustworthiness benchmark that directly tackles data leakage, cost, and adaptability**, rather than inheriting these limitations from existing static resources. Simply aggregating existing static benchmarks would only increase coverage on paper while preserving these structural problems.
>
> At the same time, our motivation for a unified evaluation framework is grounded in how generative models are actually developed and used. Although LLMs, VLMs, and text to image models operate on different modalities, they share a common computational paradigm: they take an input prompt and generate an output sample. More importantly, they face similar trustworthiness challenges, such as hallucinations and truthfulness issues, safety violations, fairness and bias, and potential privacy leaks. These are recurring patterns of failure in generative modeling, not purely modality specific concerns. Evaluating each modality in isolation fragments what we view as a fundamentally unified research problem.
>
> Our framework is therefore designed as a modular and modality aware infrastructure. We define shared trustworthiness dimensions, including truthfulness, safety, robustness, and fairness, while instantiating them in a way that respects modality specific characteristics. For example, we consider hallucination and grounding for VLMs, compositional correctness and harmful content for text to image models, and machine ethics and instruction following for LLMs. In this sense, the framework provides a coherent conceptual backbone that can be specialized to different model classes rather than forcing a single protocol on every modality.
>
> This unified and dynamic design also enables cross model insights that would be hard to obtain from separate, static benchmarks. For instance, we observe that models released by the same organization can exhibit coherent safety behavior patterns across modalities, which suggests shared alignment philosophies across product lines. Our framework reveals such patterns not by numerically ranking LLMs against text to image models, but by placing them in a common conceptual and infrastructural space, which we believe is valuable for both researchers and practitioners.
>
> To directly address your concern about comparability, our goal is not to create a single cross modality leaderboard. Instead, “unified” means shared evaluation infrastructure, consistent dimensions, and a dynamic data generation process that can be extended to new model types over time. Researchers who work on different modalities should not need to reinvent trustworthiness evaluation from scratch, nor manually stitch together heterogeneous benchmarks with incompatible protocols.

---

> ### Author Response · Authors · 2025-11-15
> **Response for Reviewer u6eR (Part III)**
>
> Regarding the concern about data quality in dynamic construction, we fully agree this is a crucial issue. Prior dynamic evaluation efforts often fall into one of two extremes. Some rely on purely programmatic construction, which typically yields high precision but limited diversity and relatively homogeneous samples. Others rely primarily on generative model based construction, which improves diversity and controllability but can introduce more noise. **TrustGen adopts a hybrid strategy that combines programmatic generation with generative model augmentation.** This design, detailed in §2.1 across our three modules, allows us to expand the evaluation set adaptively while enforcing structural constraints and consistency checks that maintain quality.
>
> We do not claim that TrustGen is perfect, but **human evaluations reported in Appendix H, J, and K show that our generated test data achieves satisfactory reliability and validity in practice**. We provide detailed inter annotator agreement statistics and qualitative analyses of data quality. In our camera ready version, we will surface these validation results more prominently in the main paper to better substantiate and clarify our design choices.
>
> In summary, **our motivation is to address the core limitations of static, fragmented trustworthiness benchmarks by proposing a dynamic and unified evaluation framework** that treats generative models, regardless of modality, as part of a common trust challenge, while still respecting their differences in a principled and carefully validated way.
>
> [1] Xu, Ruijie, et al. "Benchmarking benchmark leakage in large language models." arXiv preprint arXiv:2404.18824 (2024).
>
> [2] Zhu, Qin, et al. "Inference time decontamination: Reusing leaked benchmarks for large language model evaluation." Findings of the Association for Computational Linguistics: EMNLP 2024. 2024.
>
> [3] Chen, Simin, et al. "Recent advances in large language model benchmarks against data contamination: From static to dynamic evaluation." arXiv preprint arXiv:2502.17521 (2025).
>
> [4] Zhu, Kaijie, et al. "DyVal: Dynamic Evaluation of Large Language Models for Reasoning Tasks." The Twelfth International Conference on Learning Representations.
>
> [5] Zhu, Kaijie, et al. "DyVal 2: Dynamic evaluation of large language models by meta probing agents." arXiv preprint arXiv:2402.14865 3 (2024).
>
> [6] Wang, Siyuan, et al. "Benchmark self evolving: A multi agent framework for dynamic LLM evaluation." Proceedings of the 31st International Conference on Computational Linguistics. 2025.
>
> [7] Li, Jiatong, et al. "DynaEval: A Dynamic Interaction based Evaluation Framework for Assessing LLMs in Real world Scenarios."

---

> ### Author Response · Authors · 2025-11-15
> **Response for Reviewer u6eR (Part IV)**
>
> ### Q2: Paper structure and dimension definitions
>
> **Your Concern:** The structure of the paper is somewhat unreasonable for a benchmarking study. The main body should present detailed information about dimensions, datasets (size, case examples), and metrics. Additionally, Table 1 shows that evaluation dimensions for different model types appear to be defined independently. Why should the dimensions of trustworthiness vary for each model? For instance, if LLMs can generate toxic output, why would VLMs be exempt from this concern? More rigorous definitions for trustworthiness dimensions are needed. Safety and robustness are often discussed synonymously, so their distinct roles need justification. Critical threats like backdoor attacks and watermarking are not discussed.
>
> **Our Response:**
>
> We sincerely thank you for these constructive comments and completely understand your concern about the paper's structure.
>
> **Regarding the organization of content:** We must confess that we faced a genuinely difficult dilemma during manuscript preparation. The full version of TrustGen originally exceeded **200 pages** before any trimming, and we found ourselves in an impossible situation: either include all technical details in the main text (making it unreadable and unfair to reviewers), or carefully reorganize the content to balance completeness with accessibility. We chose the latter approach and made our organizational strategy explicit in **Lines 151-154 (Draft Organization)**, where we explain how different content types are distributed across sections.
>
> To directly address your request for **detailed information about dimensions, datasets, and metrics**, we want to clarify that this information is **comprehensively documented** in the appendix, it was not omitted, but rather strategically placed to maintain readability:
>
> - **Benchmark details** (including rigorous dimension definitions, evaluation metrics, and construction methodologies) → **Appendix D**
> - **Dataset examples** (concrete cases for each dimension across all model types) → **Appendix P**
> - **Dataset sizes and statistics** → We have now added **Appendix O (Table 37)** specifically in response to feedback, providing complete dataset scale information for every dimension and model type. (Note: The dataset is also publicly available on Hugging Face, with the link provided directly below the abstract.)
>
> We hope this roadmap helps you navigate to the specific technical details you need. If any information remains unclear or difficult to locate, please let us know, we would be very happy to point you to the exact sections.
>
> **Regarding your question "why should the dimensions of trustworthiness vary for each model?":** Our design is **modality-aware but conceptually unified**. For example, VLMs introduce a visual input modality but still generate textual outputs. Thus, the **toxicity dimension**, which measures textual harmfulness, largely evaluates the language generation component rather than the visual reasoning part, making it somewhat redundant for VLMs. Conversely, dimensions like **jailbreak robustness** are highly relevant for VLMs, since visual inputs can trigger new forms of prompt exploitation. We describe this rationale explicitly in **§2.2 (first two paragraphs)**, where each dimension is contextualized to the model type while maintaining alignment with the overarching taxonomy.
>
> **On your point that safety and robustness are often discussed synonymously:** We appreciate this observation and have clarified the distinction in **Section 2.2** of the revised text. In TrustGen, **robustness** refers to the model's stability under input-level perturbations or adversarial noise, whereas **safety** focuses on content-level harmfulness and ethical constraints. These two aspects are orthogonal in generative evaluation and therefore treated separately.
>
> **As for critical threats such as backdoor attacks and watermarking:** We acknowledge that the submitted version omitted these due to space limitations. We have now **added a detailed discussion in Appendix F3.5**. This new section summarizes relevant prior work and outlines our ongoing plan for incorporating backdoor- and watermark-oriented evaluations into the benchmark. We will additionally surface these key points in the main paper to make the scope of coverage more explicit.
>
> We genuinely thank you for these suggestions, they have helped us realize that our organizational strategy, while necessary, may not have been sufficiently signposted for readers. If there remain specific aspects or references you believe we should include, we would be very happy to incorporate them.

---

> ### Author Response · Authors · 2025-11-15
> **Response for Reviewer u6eR (Part V)**
>
> All relevant modifications have been marked in blue in the revised version.
>
> We hope our responses have adequately addressed your concerns. In light of these clarifications and revisions, we would be sincerely grateful if you could consider revisiting your evaluation of our manuscript. Your feedback has been invaluable in helping us improve the paper's clarity and completeness. If you have any further questions or suggestions, please feel free to share them. Thank you again for your careful review!

---

> ### Author Response · Authors · 2025-11-26
> **Gentle Reminder Regarding Our Response to Your Comments**
>
> Dear Reviewer,
>
> Thank you very much for your valuable time and thoughtful comments. We completely understand that you may have a busy schedule. As the discussion period is now less than one week remaining, we would be truly grateful if you could kindly take a moment to review our responses and let us know whether they adequately address your concerns.
>
> We sincerely appreciate your time and consideration.

---

> > ### Comment · Reviewer_u6eR · 2025-11-28
> >
> > Thank you for your response. I agree on the need for a dynamic benchmark, though I still have some reservations regarding the reliability of TrustGen. One suggestion would be to weight the evaluation metrics according to data quality.
> >
> > On Q2, I would like to stress the importance of including benchmark details and dataset statistics in the main text. I think most researchers do not have sufficient time to read the full appendix. Providing some high-level summaries in the main paper, along with clear pointers to the relevant sections of the appendix, would improve accessibility.
> >
> > Moreover, I encourage the authors to discuss the limitations of TrustGen in the main text as well, e.g., potential evaluation pitfalls introduced by TrustGen and the dimensions that this work does not cover.
> >
> > Overall, I appreciate the authors’ efforts, and I am willing to raise my score to 6.

---

> > > ### Author Response · Authors · 2025-11-28
> > > **Thanks for your reply**
> > >
> > > Thank you very much for your kind recognition. Although the current official policy resets all scores back to their initial state after review, we truly appreciate your response; it is a great encouragement to us!

---

### Official Review · Reviewer_HR7f · 2025-11-02

**Soundness:** 2
**Presentation:** 2
**Contribution:** 2
**Rating:** 4
**Confidence:** 3

**Summary:**

This paper presents TrustGen, a dynamic and modular benchmarking platform designed to evaluate the trustworthiness of Generative Foundation Models (GenFMs)—including large language models (LLMs), text-to-image (T2I) systems, and vision-language models (VLMs). The proposed framework standardizes trust evaluation through a unified taxonomy of over 25 fine-grained dimensions covering truthfulness, safety, fairness, robustness, privacy, and ethics. TrustGen consists of three core modules—Metadata Curator, Test Case Builder, and Contextual Variator—that enable dynamic dataset generation and adaptive evaluations. The authors benchmarked 39 generative models and reported four general insights, such as narrowing trust gaps among top models and competitive performance from open-source systems. The paper claims that TrustGen fills the gap in fragmented and static evaluations, offering a unified, extensible platform for trustworthiness assessment.

**Strengths:**

1. Timely and important topic – The paper addresses a highly relevant problem in the current AI landscape: how to evaluate the trustworthiness of diverse generative models. As GenFMs are increasingly applied in sensitive domains, systematic evaluation frameworks like TrustGen are much needed.

2. Comprehensive coverage and modular design – The proposed benchmark encompasses multiple modalities (text, vision, and multimodal) and integrates numerous trust-related dimensions. The three-module architecture (Metadata Curator, Test Case Builder, and Contextual Variator) is thoughtfully designed and ensures flexibility and extensibility.

3. Extensive empirical evaluation – The authors evaluate a large set of mainstream models (LLMs, T2I, and VLMs) and provide cross-model analysis. The large-scale experimentation adds credibility to the platform’s applicability and potential impact.

**Weaknesses:**

1. Unclear motivation for mixed cross-model evaluation – While the authors emphasize that existing works evaluate model families in isolation (LLMs or T2I), the paper does not convincingly justify why evaluating different modalities together is necessary or what unique insights such a unified evaluation yields. The presented findings do not clearly demonstrate benefits that could not be obtained through separate, modality-specific assessments. Thus, the key motivation and novelty of the “cross-family” benchmarking remain insufficiently supported.

2. Lack of surprising or deep insights – The four reported insights (e.g., narrowing trust gap, open-source competitiveness, correlation between trust and utility) are largely expected and do not provide new conceptual understanding of GenFM trustworthiness. The results read more as descriptive summaries rather than analytical insights that could guide future model design or policy development.

3. Inconsistent evaluation dimensions across modalities (Table 1) – Table 1 shows that different model categories (LLMs, T2I, VLMs) are evaluated under non-uniform trustworthiness dimensions and metrics, which undermines the claim of a unified benchmark. This inconsistency weakens the argument for the necessity of a cross-modal evaluation platform and raises questions about whether TrustGen truly enables comparable assessments across model families

**Questions:**

see the weakness

---

> ### Author Response · Authors · 2025-11-15
> **Response for Reviewer HR7f (Part I)**
>
> Dear Reviewer HR7f,
>
> Thank you very much for your thoughtful comments and constructive feedback. We greatly appreciate the time and effort you have invested in reviewing our work. Below, we address each of your concerns in detail.
>
> ### Q1: Unclear motivation for mixed cross-model evaluation
>
> **Your Concern:** The motivation for evaluating different model types (LLM, VLM, T2I) together is unclear, and the differences in evaluation dimensions across model types raise questions about comparability.
>
> **Our Response:**
>
> We appreciate this question, it goes to the heart of our design philosophy. Let us be direct about our motivation: **all generative models, regardless of modality, share fundamental trust challenges that demand unified evaluation infrastructure.**
>
> **(1) They are fundamentally the same class of models.** While LLMs, VLMs, and text-to-image models differ in modality, they share a common computational paradigm: they take an input prompt and generate an output sample. More critically, they face **the similar trust failures**—truthfulness, safety violations, fairness issues,  privacy leaks and so on. These are not modality-specific problems; they are inherent to generative modeling itself. Evaluating them separately would artificially fragment what is essentially a unified research challenge.
>
> **(2) Unified infrastructure enables, not restricts.** Our framework is explicitly designed to be **modular and modality-aware**. We define shared trustworthiness dimensions (truthfulness, safety, robustness, fairness, etc.), but instantiate them appropriately for each model type, for example, hallucination evaluation for VLMs, compositional correctness for text-to-image models, and machine ethics for LLMs. This is not conceptual confusion; it is **deliberate flexibility**. The framework preserves conceptual coherence while respecting modality-specific realities, and enables plug-and-play extension to future generative models.
>
> **(3) Cross-model insights emerge only through unified evaluation.** Evaluating these models together reveals patterns invisible in isolation. For example, we observe that generative models from the same organization (e.g., OpenAI) exhibit **remarkably coherent safety behaviors across modalities** (Figures 2, 3, 4), not because we force numerical comparison, but because unified evaluation exposes shared alignment philosophies. These insights have direct implications for understanding how organizational practices shape model trustworthiness across product lines.
>
> **(4) The community needs this.** Here is the practical reality: researchers currently must navigate dozens of fragmented, modality-specific benchmarks, each with different metrics, protocols, and infrastructures. This is inefficient and unsustainable. **Our goal is simple: make trustworthiness evaluation accessible to everyone working on generative models.** Whether you build LLMs, VLMs, or text-to-image systems, you should be able to evaluate, diagnose, and improve your model's trustworthiness using a single, evolving platform, not reinvent evaluation infrastructure from scratch.
>
> To directly address your concern about comparability: **we never intended to rank LLMs against text-to-image models in the same leaderboard.** "Unified" means shared evaluation infrastructure and conceptual framework, not cross-modality performance scoreboards.
>
> We hope this clarifies our motivation. If you believe our design choices remain unclear or problematic, we would genuinely appreciate specific feedback on what alternative approach you would recommend.

---

> ### Author Response · Authors · 2025-11-15
> **Response for Reviewer HR7f (Part II)**
>
> ### Q2: Lack of surprising or deep insights
>
> **Your Concern:** The findings presented in the paper lack surprising or deep insights that would make the work more impactful.
>
> **Our Response:**
>
> We sincerely thank you for this thoughtful comment and fully understand the expectation for findings that feel "new" or "surprising." We share that aspiration too, but in a benchmarking study, the core value lies not in dramatic novelty, but in **turning intuition into reproducible, quantifiable, and actionable evidence**. What may seem unsurprising on the surface often becomes meaningful when validated rigorously across dozens of models, modalities, and trustworthiness dimensions under a unified evaluation framework.
>
> In our case, the four high-level findings summarized in the main text were selected because they represent **foundational, evidence-backed truths** about the current generation of foundation models, the kind of results that help the community build on a stable empirical base rather than speculation. **We explicitly noted in Lines 108-109 that these are high-level insights, with more detailed findings documented in the appendix.** Their value lies in their **clarity, generality, and practical interpretability**, as outlined below:
>
> - **(1) State-of-the-art models still face trustworthiness bottlenecks.**
>    This finding establishes, for the first time, a *measurable ceiling* for model trustworthiness under controlled, multi-modal evaluation. It provides both developers and policymakers with clear, evidence-based targets for improvement, identifying where progress remains limited rather than relying on anecdotal impressions.
> - **(2) Open-source models are catching up with proprietary ones.**
>    This result carries strong strategic and social implications. It challenges the long-held assumption that "closed models are safer," and demonstrates that with proper alignment and safeguard strategies, open-source models can achieve or even surpass proprietary systems. This evidence directly informs fairer adoption and governance decisions across the ecosystem.
> - **(3) The trustworthiness gap among top-tier models is narrowing.**
>    This convergence reflects the diffusion of trustworthy AI practices across the industry. It shows that the community is maturing collectively, with model developers sharing best practices and aligning toward more standardized principles of responsible model design.
> - **(4) Trustworthiness creates ripple effects across dimensions.**
>    This finding provides a *methodological insight*: dimensions such as safety, helpfulness, and robustness are deeply interdependent. It cautions against optimizing one axis in isolation and supports the growing emphasis on **multi-objective alignment** and holistic trust evaluation.
>
> Taken together, these four findings form a coherent picture of where we stand as a field, not shocking, but **deeply consequential** for anyone building, auditing, or deploying foundation models responsibly. They serve as the empirical backbone on which future theoretical or mechanistic studies can build.
>
> **Beyond these high-level findings, the appendix contains a wealth of detailed insights.** To make these more accessible and to address your concern about depth, we have structured the appendix such that **for each dimension of each model type, we explicitly highlight at least one key finding/insight using underlined italics**. This results in **over 20 clearly marked, dimension-specific insights** throughout the appendix, far more than could reasonably fit in the main text without overwhelming readers. We deliberately adopted this presentation strategy (as signposted in Lines 108-109) to balance comprehensiveness with readability: the main text provides the big picture, while the appendix offers granular, actionable findings for those who want to dive deeper into specific trustworthiness aspects.

---

> ### Author Response · Authors · 2025-11-15
> **Response for Reviewer HR7f (Part III)**
>
> For example, these detailed analyses reveal richer and often counterintuitive trends:
>
> - **T2I Privacy (Appendix E.6):** Some text-to-image models exhibit markedly different leakage rates for organizational vs. individual privacy content, showing how privacy risks depend on contextual semantics.
> - **LLM Truthfulness (Appendix F.2):** Smaller models are sometimes more robust to persona and preconception sycophancy, even though larger ones are generally stronger overall, a subtle but important dynamic for model scaling research.
> - **LLM Robustness (Appendix F.5):** Models tend to perform better on annotated datasets than on open-ended prompts, exposing a critical generalization gap between structured and real-world evaluation settings.
> - **LLM Advanced AI Risk (Appendix F.8):** Larger and more advanced models outperform smaller ones in reasoning about advanced AI risks, suggesting that scaling enhances caution and reliability in sensitive reasoning tasks.
>
> These are only a few among many detailed analyses that make up the full empirical landscape of our study. We will make this connection clearer in the revised version by explicitly referencing representative appendix results in the main text, guiding readers to the wealth of insights documented throughout the appendix.
>
> In short, while our findings may not be "surprising" in a sensational sense, they are **foundational, interpretable, and actionable**, the kind of carefully verified evidence that moves a research community forward and builds the ground truth upon which future innovation rests.

---

> ### Author Response · Authors · 2025-11-15
> **Response for Reviewer HR7f (Part IV)**
>
> ### Q3: Inconsistent evaluation dimensions across modalities
>
> **Your Concern:** The evaluation dimensions appear inconsistent across different model types, which raises questions about whether TrustGen truly enables comparable assessments across model families.
>
> **Our Response:**
>
> We understand where this concern comes from, and we appreciate the opportunity to clarify. The observed differences across model types are **driven by practical reality, not arbitrary design choices**. We first establish a unified taxonomy of trust dimensions for generative models, but we only instantiate the dimensions that are actually meaningful for a given modality. For example, evaluating "Machine Ethics" on a T2I model is not meaningful, since such models do not engage in moral reasoning or text-based judgment. This is explicitly stated in §2.2, our framework defines a shared conceptual space, but not all dimensions are applicable to every model type.
>
> Our toolkit is intentionally **modular and on-demand**. Users can select the relevant axes and probes for each model; skipping an irrelevant dimension does not reduce rigor, since data generation, scoring, and probing are independent components that can be flexibly combined. This modular design reflects the reality of heterogeneous generative models rather than forcing them into a single rigid template.
>
> **As for your comment that this "raises questions about whether TrustGen truly enables comparable assessments across model families," we believe there may be a misunderstanding. We never intended** to compare fundamentally different model types (e.g., LLMs vs. T2I models) under the same metric space. "Unified" in our context refers to shared input/output paradigm, scales, and ideas of dynamic data construction, not to direct performance ranking across modalities.
>
> We sincerely thank you for raising this point; it helps us clarify that TrustGen aims to be flexible, inclusive, and faithful to the nature of generative models, **a foundation for understanding, not a scoreboard for comparison.**

---

> ### Author Response · Authors · 2025-11-15
> **Response for Reviewer HR7f (Part V)**
>
> All relevant modifications have been marked in blue in the revised version.
>
> We hope that our responses have addressed your concerns and helped clarify the key ideas and contributions of our work. We are genuinely excited about these improvements, and if you feel that our replies have satisfactorily resolved your concerns, we would be very grateful for your re-evaluate and potential positive reconsideration of our manuscript. If you have any further questions or would like additional clarifications, please do not hesitate to let us know. Your thoughtful feedback has been truly valuable in helping us refine and better present our work!

---

> ### Author Response · Authors · 2025-11-26
> **Gentle Reminder Regarding Our Response to Your Comments**
>
> Dear Reviewer,
>
> Thank you very much for your valuable time and thoughtful comments. We completely understand that you may have a busy schedule. As the discussion period is now less than one week remaining, we would be truly grateful if you could kindly take a moment to review our responses and let us know whether they adequately address your concerns.
>
> We sincerely appreciate your time and consideration.

---

### Official Review · Reviewer_uHJ4 · 2025-11-03

**Soundness:** 3
**Presentation:** 2
**Contribution:** 4
**Rating:** 6
**Confidence:** 4

**Summary:**

TrustGen introduces a dynamic, modular benchmark for assessing the trustworthiness of generative foundation models in LLM, T2I, and VLM. It standardizes evaluation via a unified taxonomy, including truthfulness, safety, fairness, robustness, privacy, machine ethics, etc. The benchmark continuously generates up-to-date test cases via three modules: Metadata Curator, Test Case Builder, and Contextual Variator. Applied to 39 models, TrustGen yields several notable findings, including strong average performance but persistent gaps in areas such as hallucination, fairness, and privacy, with some open-source systems matching the performance of frontier proprietary models.

**Strengths:**

- Comprehensive framework: built on a broad, fine-grained taxonomy and a reusable cross-modal pipeline, validated by broad empirical coverage, evaluating 39 models across modalities.

-  Dynamic data generation: Moves beyond static benchmarks by iteratively producing fresh test cases with minimal human intervention, better matching a rapidly changing threat landscape.

- Human Review: the authors provide human verification for both test case generation and result evaluation, demonstrating the reliability of using LLMs to generate cases and to act as a judge within the TrustGen framework.

-  Reusability: Public dataset link and toolkit.

**Weaknesses:**

-  Dynamic-benchmark comparability: While dynamic is a strength, it complicates leaderboard stability. Strict versioning will be needed for comparisons.

- Expository clarity requires improvement: missing details make the core contributions hard to discern.

**Questions:**

1.  Writing and emphasis: I appreciate the substantial effort required to produce this work, including task selection, data-generation pipeline design, metric comparison, and the modular framework. However, the manuscript places disproportionate emphasis on analyzing evaluation results. Unlike HELM, DecodingTrust, MultiTrust, and Safebench, a core contribution here is dynamic data generation and adaptive evaluation. This aspect deserves more space and clearer exposition so readers can understand what “dynamic” concretely entails.

    -   Please expand the section 2.1, or align each subsequent evaluation sections to the three key modules in Figure 1 and provide the corresponding data generation pipeline. The appendix only briefly describes the dataset composition，lacks concrete details on how each component is constructed, as well as dataset sizes.

    -   For each dimension, the manuscript does not clearly clarify differences from prior work.

2. Benchmark fairness：how is fairness ensured for dynamic datasets?

3. Future work: I strongly recommend adding a dedicated future work section to outline follow-up plans, such as introducing additional modalities or model families, e.g., audio LLMs and text-to-video models. For a benchmarking platform that required substantial effort to build, sustained iteration and updates are essential, without an ongoing roadmap, its long-term value will be limited.

4. The paper covers many trustworthiness aspects, and different sections were likely written by different authors. Please align the writing template across dimensions.

    -   In the appendix, section sub-titles are inconsistent, e.g., many dimensions omit a Dynamic Dataset subsection.

    -   In F2.1, what is the purpose of introducing standard TrustLLM datasets and comparing them with dynamic datasets? It seems that the model achieves higher acc on dynamic datasets compared to TrustLLM. May I infer that dynamic datasets are less capable of fully revealing a model’s internal safety risks?

5. Minor: The model order in the main-text tables is unusual. For the GPT series for instance, the ordering is neither chronological nor by capability.

These questions require little to no additional experimentation. I would be pleased to revise my score if the authors provide reasonable responses.

---

> ### Author Response · Authors · 2025-11-15
> **Response for Reviewer uHJ4 (Part I)**
>
> Dear Reviewer uHJ4,
>
> Thank you very much for your encouraging and constructive feedback. We are delighted that you found our work valuable, and we greatly appreciate the detailed suggestions you have provided. Below, we address each of your points carefully.
>
> ### Q1: Expand Section 2.1 and provide detailed data generation pipelines
>
> **Your Suggestion:** Please expand Section 2.1, or align each subsequent evaluation section to the three key modules in Figure 1 and provide the corresponding data generation pipeline. The appendix only briefly describes the dataset composition and lacks concrete details on how each component is constructed, as well as dataset sizes.
>
> **Our Response:**
>
> We sincerely thank you for these valuable comments. We fully agree that Section 2.1 and the description of our data generation modules are critical to understanding TrustGen.
>
> Regarding your suggestion to "expand Section 2.1 or align subsequent evaluation sections with the three key modules in Figure 1," we would like to clarify that **Table 1 already summarizes how each module is instantiated across different model types and trust dimensions**. Each module, data generation, probing, and scoring, is implemented differently depending on the modality, and these correspondences are explicitly presented in the table.
>
> On your remark that "the appendix only briefly describes the dataset composition," we believe there may be a misunderstanding. The construction details are not limited to the short "Dynamic Dataset" paragraphs; they are also elaborated in each dimension's **"Benchmark Setting"** subsection, which provides the generation logic, data filtering steps, and validation settings. Taken together, these two parts describe the full data-generation pipeline. Moreover, **the dataset metadata and statistics are released on Hugging Face** (linked directly below the abstract). To make this clearer, we have now **added explicit dataset statistics to Appendix O (Table 37), so readers can directly trace every dataset component**.
>
> ### Q2: Clarify differences from prior work for each dimension
>
> **Your Concern:** For each dimension, the manuscript does not clearly clarify differences from prior work.
>
> **Our Response:**
>
> Thanks for your suggestions! We would like to clarify two cases separately:
>
> First, if this refers to benchmark-level comparisons, we have already provided an extensive analysis in **Appendix C**, where TrustGen is systematically compared against over 40 existing benchmarks covering LLM, T2I, and VLM models. We believe this already provides a comprehensive mapping between our framework and prior benchmark efforts.
>
> Second, if your comment instead refers to broader related works (beyond benchmarking), we sincerely apologize and hope you can understand our dilemma. In an earlier draft, we had included more than 1,500 citations across dimensions, covering every subtopic of trustworthy generative modeling. However, this made the paper read almost like a survey, and the total length exceeded 200 pages before trimming. We found ourselves in a genuinely difficult situation: either remove hundreds of citations (and risk appearing incomplete), or retain them and risk producing a manuscript that would be impossible to review. We reluctantly chose the former to preserve readability and fairness to reviewers.
>
> We fully acknowledge that this compromise may have given the impression that some distinctions from prior work were underemphasized. If you believe that any important related works have been inadvertently omitted, please kindly let us know, we would be very happy to include them in the revision!

---

> ### Author Response · Authors · 2025-11-15
> **Response for Reviewer uHJ4 (Part II)**
>
> ### Q3: How is fairness ensured for dynamic datasets?
>
> **Your Question:** How is fairness ensured for dynamic datasets in benchmark evaluation?
>
> **Our Response:**
>
> We sincerely thank you for this very insightful question. We fully agree that fairness is a critical concern for any dynamic benchmarking framework, and we have carefully considered this issue when designing TrustGen.
>
> TrustGen serves two complementary purposes: (1) **Public leaderboard evaluation**, and (2) **Individual dynamic evaluation.**
>
> For the **public leaderboard**, fairness is ensured through version-controlled benchmark releases. Each benchmark version is publicly maintained and fixed once released, meaning all models evaluated under the same version are tested on identical data. When the benchmark is updated, a new version number is issued, and results are reported separately across versions. This design allows continuous improvement of the benchmark while ensuring that comparisons remain fair within each release cycle.
>
> For **individual dynamic evaluation**, the goal is different: users can employ the TrustGen toolkit to locally generate customized test sets to identify weaknesses in their own models. These evaluations are not meant for leaderboard comparison but rather for self-assessment and model diagnosis. This flexibility is precisely what makes TrustGen practical, users can explore new trust dimensions or tailor evaluations to their use cases without compromising the fairness of the public benchmark.
>
> To make this clearer, we have now added a dedicated section on **Fairness Ensurance** in **Appendix R**, where we describe the version-control protocol and the dual-role design of TrustGen in detail. We sincerely thank you for this helpful suggestion, it has made our benchmark presentation more transparent and complete.
>
> ### Q4: Add a dedicated future work section
>
> **Your Suggestion:** I strongly recommend adding a dedicated future work section to outline follow-up plans, such as introducing additional modalities or model families (e.g., audio LLMs and text-to-video models). For a benchmarking platform that required substantial effort to build, sustained iteration and updates are essential; without an ongoing roadmap, its long-term value will be limited.
>
> **Our Response:**
>
> We sincerely thank you for this excellent suggestion. We completely agree that outlining a clear roadmap and future extensions is essential for a benchmarking platform like TrustGen, whose long-term value depends on sustained updates and inclusion of emerging generative modalities.
>
> In earlier versions of the manuscript, we indeed included a detailed discussion on future work, but it was later removed due to strict page limitations during submission. Following your recommendation, we have now restored and substantially expanded this discussion in **Appendix S**, under a new section titled **"Other Generative Models."**
>
> This section presents our planned extensions to **Any-to-Any Models**, **Video Generative Models**, **Audio Generative Models**, and **Generative Agents**, along with **100+ newly added references** covering representative works in these areas. Together, they outline our roadmap for extending TrustGen to broader generative model families and for maintaining it as a continuously evolving, community-driven benchmark.
>
> We truly appreciate this suggestion, it has helped us make the long-term vision of TrustGen much clearer and more forward-looking!

---

> ### Author Response · Authors · 2025-11-15
> **Response for Reviewer uHJ4 (Part III)**
>
> ### Q5: Purpose and capability of dynamic datasets
>
> **Your Observation:**
>  In F2.1, what is the purpose of introducing standard *TrustLLM* datasets and comparing them with dynamic datasets? It seems that the model achieves higher accuracy on dynamic datasets compared to *TrustLLM*. May I infer that dynamic datasets are less capable of fully revealing a model’s internal safety risks?
>
> **Our Response:**
>
> We thank the reviewer for this detailed and constructive question. We include the standard *TrustLLM* dataset because our framework evaluates models using two complementary data sources:
>  (1) a **static dataset pool**, which randomly samples from existing benchmark datasets, and
>  (2) a **dynamic dataset** generated by *TrustGen*, which continuously collects newly observed data from real-world sources.
>
> We acknowledge that the dynamically generated datasets may not always be as carefully curated or as challenging as highly optimized static benchmarks such as *TrustLLM*. However, the **core motivation** of dynamic evaluation is not to reproduce the same level of static difficulty, but to enable an *open-ended and continuously evolving challenge space* that can adapt to new risks, topics, and societal contexts. Over time, as *TrustGen* accumulates and refreshes data, it can surface a *broader and more diverse* set of challenges than any fixed dataset, effectively expanding the scope of safety risk exposure beyond what static evaluations can reveal.
>
> Therefore, the slightly higher model accuracy observed on the current dynamic dataset mainly reflects the particular data distribution at the time of collection, rather than a fundamental limitation in its ability to reveal safety risks. With each new release, the dataset evolves, incorporating new types of prompts and real-world phenomena. We believe this long-term adaptability, rather than one-time difficulty, is what ultimately strengthens the evaluation of model trustworthiness.
>
> We have added this clarification to **Appendix F2.1**, emphasizing that *TrustGen*’s dynamic datasets are designed to complement static benchmarks by ensuring *breadth*, *timeliness*, and *sustained novelty* in model evaluation.
>
> ### Q6: Inconsistent sub-titles in the appendix
>
> **Your Observation:** In the appendix, section sub-titles are inconsistent, for example, many dimensions omit a Dynamic Dataset subsection.
>
> **Our Response:**
>
> We sincerely thank you for this careful observation. Following your suggestion, we have made several consistency improvements across the appendix to ensure clarity and uniformity. Specifically, we have:
>
> (1) Standardized the capitalization and formatting of all sub-titles, and
>
> (2) Unified section naming conventions across different model types and dimensions, for example, each T2I-related dimension (such as Truthfulness) now explicitly includes both Benchmark Settings and Dynamic Dataset subsections.
>
> For certain cases where a Dynamic Dataset subsection was not originally included, such as LLM Toxicity, this was intentional, as the dataset construction in that dimension primarily relies on the Jailbreak data pipeline, with the difference lying only in evaluation methodology. We have clarified this relationship directly within the Toxicity section to avoid any confusion.

---

> ### Author Response · Authors · 2025-11-15
> **Response for Reviewer uHJ4 (Part IV)**
>
> ### Q7: Unusual model ordering in tables
>
> **Your Observation (Minor):** The model order in the main-text tables is unusual. For the GPT series, for instance, the ordering is neither chronological nor by capability.
>
> **Our Response:**
>
> We sincerely thank you for pointing this out. We have carefully checked all tables and revised the model ordering to follow a consistent release time sequence within each organization (for the GPT series). We greatly appreciate your attention to detail!

---

> ### Author Response · Authors · 2025-11-15
> **Response for Reviewer uHJ4 (Part V)**
>
> All relevant modifications have been marked in blue in the revised version.
>
> We truly appreciate your thoughtful comments and the encouragement you have given us. Your feedback has helped us see our work from new perspectives and make it stronger. We hope our responses have addressed your concerns, and we would be very happy to clarify anything further if needed. Thank you again for the time and care you have put into reviewing our paper, it really means a lot to us!

---

> ### Author Response · Authors · 2025-11-26
> **Gentle Reminder Regarding Our Response to Your Comments**
>
> Dear Reviewer,
>
> Thank you very much for your valuable time and thoughtful comments. We completely understand that you may have a busy schedule. As the discussion period is now less than one week remaining, we would be truly grateful if you could kindly take a moment to review our responses and let us know whether they adequately address your concerns.
>
> We sincerely appreciate your time and consideration.

---

### Meta-Review · Area_Chair_8ETf · 2026-01-17

**Summary:**

TrustGen introduces a dynamic, modular benchmark for assessing the trustworthiness of generative foundation models in LLM, T2I, and VLM.

Reviewer expressed following main concerns.
- Presentation clarity
- Unclear motivation for cross-modal evaluation
- Inconsistent evaluation dimensions across modalities
- Advantage over existing benchmarks.

**Reviewer Concerns:**

Authors revised the paper in the rebuttal period
- Expanded section 2.1 and provided details about data generation pipeline. Clarified difference from prior work and explained fairness metrics.
- Authors explained the motivations behind cross-modal evaluations and evaluation dimensions.
- Authors explained the merits of the proposed benchmark over existing ones.

**Reviewer Scores:**

Reviewer scores were 6, 4, 4
One reviewer raised their score from 4 to 6 during discussion phase.

The author responses seem reasonable to me and might have led to increase in the overall scores.

I have sent a separate message about potential violation of double-blind submission policy.
On technical merits, I am leaning toward accept, but on the strict policy adopted by ICLR in other cases, I think this paper is a candidate for desk reject. For these reasons I am not sure what is the right decision and would like SAC or PCs to take a look.

---

### Decision · Program_Chairs · 2026-01-26

Accept (Poster)